

# Tsunami Risk and Alert Perceptions in Five Municipalities bordering the French Mediterranean Basin

Johnny DOUVINET[1,2], Noé CARLES[1], Pierre FOULQUIER[1], Matthieu PEROCHE[3]

[1] UMR ESPACE 7300 CNRS, Avignon University, Avignon, F-84000, FRANCE
[2] French University Institute, IUF, Paris, F-75008, FRANCE
[3] UPR GEAD, Paul Valery Montpellier III University, F-34000, FRANCE

*Correspondence to*: Johnny DOUVINET (johnny.douvinet@univ-avignon.fr)

**Abstract.** Since the major tsunami that occurred in 2004, many studies have dealt with the evacuations, hazard mapping and awareness-actions, rather than on the perceived tsunami risk nor on alert perceptions declared locally by the population. In this study, we analyzed a sample of 750 answers to a face-to-face questionnaire, gathered from residents or workers in 5 densely-urbanized municipalities (e.g., Bandol, Bastia, Cannes, Six-Fours-Les-Plages, Sanary-sur-Mer), likely to be hit by a tsunami and bordering the French Mediterranean basin. Results first confirmed the tendency to underestimate the tsunami risk, as only
15.6% identify tsunami as a risk. However, 48.7% declare they should take protective actions if they feel ground-shaking on the seafront, and even 65.3% if they also see a anormal sea movement. In contrast, the efficacy of alerting tools and the actors who can alert them are overestimated, as 44.7% of the respondents think they should be alerted by sirens and 11.7% by SMS, while such tools are not systematically present and rarely cover tsunami evacuation zones. And only 29.4% correctly identify the official alert senders: mayors or prefectures. In contrast, 55.7% declare they go high ground if they receive such instructions
in one Alert SMS. The age, gender, residency status or location of the respondents explain a few differences in the collected data. However, relationships between tsunami risk and alert perceptions appear statistically not corelated. All the knowledge produced in this study finally might help the municipalities further develop awareness-actions and information on the tsunami, and inform what strategy they may apply in a short future to better increase the tsunami preparedness.
**Key-Words:** tsunami; perception; coastal population; Mediterranean basin; France.

# 1. Introduction

A tsunami is a large and powerful series of ocean waves, caused by underwater disturbances such as an earthquake, a volcanic eruption, or a landslide that causes sudden vertical changes in the seafloor, which in turn displace a large volume of water from its position of equilibrium to a new position of rise or depression (Zhang et al., 2009). The 2004 tsunami that occurred in the Indian Ocean (inducing 230,000 victims, including 90,000 in the Banda Aceh area of Indonesia) raised a global awareness of





this risk (Leone & Paris, 2011; Péroche, 2016; Sun & Sun, 2019; Chen et al., 2022). Good practice guides have encouraged researchers to increase knowledge on tsunamis, to improve prevention for the concerned population and to contribute to evacuation planning and hazard mapping (Scheer et al., 2011; UNESCO/IOC, 2017). Combined with effective alerts and awareness actions, the evacuation can minimize the number of deaths. During the Tōhoku tsunami that occurred on March 11 2011, in Japan, 582,000 people (*i.e.,* 96% of the exposed individuals) were evacuated in time thanks to alerts diffused in 5

minutes. However, the increase in the population living on coastal regions (Ong et al., 2023) and the tsunamis' low frequency of occurrence (Cugliari et al., 2022a) generate a limited knowledge of risk, by citizens, local authorities, journalists, etc.

In comparison with other basins overwide, coastal areas bordering the French Mediterranean basin seem slightly exposed to tsunamis, due to the gentle number of deaths registered during past events and a very frquency (Courteau, 2017; Terrier et al., 2017; Cerase et al., 2019). According to Filippini et al., 2020, out of 167 events recorded between 1564 and 2004, only 18 real

tsunamis are reported to have really endangered such areas. However, historical sources and the tsunamis catalog are not always reliable (Terrier et al., 2017), and the recent simulations produced by Filippini et al., (2020) confirmed two scenarios at risk. On one hand, a regional tsunami may cross from the Algerian coast to the French areas between 74 to 90 minutes, with waves diffusion over a speed of hundred km.h-1. The Boumerdès earthquake that occurred in Algeria on May, 21, 2003, with a magnitude (Mw) of 6.9, illustrates such a possibility. During this event, tsunami waves were observed along the entire

Spanish and French Mediterranean coast and they were recorded by many tide gauges, even if along the French coastal areas, the run-up had not exceeded 10 centimeters (Tinti et al., 2005). A field survey showed that 8 marinas located along the eastern part (namely Côte d'Azur) experienced significant drops in sea level (50 cm to 1.5 m), basin drainage, strong eddies and currents, damaged boats, and displacement of dead bodies, all of which compatible with harbor resonance phenomena (Terrier et al., 2017). The effects on the Côte d'Azur were observed 76 minutes after the earthquake (BRGM, 2009). On the other hand,

local tsunamis could affect the coastal areas bordering the French Mediterranean basin in quick time duration, especially from the Ligurian Sea (Italy). A local tsunami could reach the Alpes-Maritimes in a duration estimated between 2 (Nice) to 7 (Cannes) minutes, in accordance with the numerical simulations (Filippini et al., 2020), thus reducing the time required for alerting and bringing the population to safety. Although of low amplitude, with modeled waves of the order of a meter, local tsunamis could cause loss of life, particularly on the crowded beaches during the peak of the summer period, due to the water

velocity, the ebb and flow phenomena, the floating debris as well as the effects of crowd movements generated. The Ligurian Sea earthquake of 23 February 1887, with a magnitude of 6.5., illustrates this other possibility. During the 1887 event, inundation was observed in Antibes, and after the first tremor, the sea suddenly receded by about one meter, leaving fishing boats and fish on the sand. Then a two-meter-high wave covered the beaches. Furthermore, the sudden breakage at two separate points of the submarine telegraph cable installed between Antibes and Corsica led to the assumption that the variable amplitude

of this tsunami was amplified by landslides, mobilizing recent sedimentary deposits (Tinti et al., 2015).

Furthermore, within the 187 municipalities likely to be hit by a tsunami in the 3 regions bordering the French Mediterranean basin, the tsunami evacuation zone covers 20,700 hectares, and possibly accommodates 163,000 residents overnight (according to the French National Institute of Statistics and Economic Studies; INSEE, 2021), and 830,000 users of beaches during the



peak of the summer season (Carles et al, 2023). By comparing the variations of the population and employment over the period 2010-2015, the average annual growth of 98% of the Mediterranean coastal municipalities is positive (INSEE, 2019). Added to the benefits from a regular changing population, thanks to geographical and heritage assets, the French Mediterranean basin also attracts many tourists (e.g., 139 million overnight stays in 2017). This population inevitably results in an increase in the human exposure to natural risks, including the tsunami risk.

Evaluating in these areas the tsunami risk and alert perceptions is thus fundamental for protecting lives and defining awareness-actions. Risk perception is based on a multidimensional approach aimed at investigating the way individual and social factors shape intuitive risk judgments on which the majority of citizens rely on (Slovic, 1987; Cugliari et al., 2022a). We assume in this study to reduce the perception to the reception of a stimulus by one of our senses, while the cognition corresponds to the translation of this stimulus by a person through individual filters such as experience, history or values (Tunstall, 2000). The perception then captures immediate realities by relying on sensory mechanisms (our five senses), and is intrinsically individual.

Knowing how people think reacting and how they represent the tsunami becomes salient as there can be dire consequences if action is not taken right away in case of tsunamis, where some additional seconds can be a matter of life and/or death (Wood et al., 2017; Weathernews Inc., 2011). Tsunamis can furthermore surprise these populations located thousands of kilometers away from the initiation area (Cerase et al., 2019), without feeling the ground-shaking. However, there is a glaring gap in the number of surveys carried out in the coastal areas bordering the French Mediterranean basin (Courteau, 2017, 2019).

In light of the limitations mentioned above, we deployed a face-to-face survey, using a questionnaire adapted from Liotard et al., 2017 and Cerase et al. (2019), consisting of 5 sections and 18 questions. Data collection had 750 answers of people living or working in the tsunami evacuation zones of cities bordering the French Mediterranean basin (Bandol, Bastia, Cannes, La Ciotat & Six-Fours-Plages). 150 answers were compiled in 5 municipalities, characterized by various densities and configurations, to perform comparative analysis. The remainder of this paper has been organized as follows: Section 2 presents

the state-of-the-art on both tsunami risk and alert perceptions, distinguished but strongly interconnected. Section 3 explains data collected and methods applied. Section 4 identifies the main trends statistically obtained. Section 5 discusses these results regarding the current French alerting process, the scenarios feared by the authorities, and the evacuation management.

## 2. Literature review

In this section, we review the related works of literature on the elements that influenced the tsunami risk and alert perceptions,
through three common points of view: 1) definitions and main features; 2) an overview of previous results obtained worldwide; 3) a synthesis of the results obtained in the western Mediterranean areas. It allows us to define our research objectives.



### 2.1. The tsunami risk perception

### 2.1.1 Definitions and main features

Defined as the process of collecting, selecting, or interpreting signals about the impacts of tsunamis, the tsunami risk perception
is influenced by numerous social and cultural factors (Slovic, 1980). Stimuli can refer to direct or indirect senses (e.g. ground-shaking, witnessing waves or floods), excluding past experiences (e.g., reading about natural disasters in newspapers), history or values (Slovic, 1987). Renn and Rohrmann (2000) have proposed a synthetic framework, Cerase et al. (2019) and Cugliari et al. (2022a) used for measuring risk perception. Four levels are distinguished: 1) the collective and individual heuristics that individuals apply during the process of forming their judgements; 2) the knowledge-based (cognitive) and the emotional-based
(affective) factors; 3) the political institutions that people and groups associate with the cause of the risk, or risk itself; 4) the cultural factors that govern or co-determine many of the lower levels of influence. Although the lead-time for the evacuation differs related to the origin of the hazard location and the geographical configurations, current processes exist for these four levels (Makinoshima et al., 2020), especially sensing threats and needs of information to reduce time for decision-making or to take protective actions. Many studies additionally suggested that the existence of knowledge and culture linking tsunamis
is essential to interpret signals (Mcadoo et al., 2006; Gaillard et al., 2008), but it clearly refers to the representation. In any case, the higher the degree of the perceived tsunami risk, the higher the preparation and the mitigation (Goto et al., 2012), and the application of preventive measures the population should take when a tsunami happens (Ong et al., 2023).

### 2.1.2 A gentle perceived tsunami risk in the western part of the Mediterranean basin

Previous surveys carried out in the western part of the overall Mediterranean basin have already confirmed the weak tsunami
risk perception. In the EU project ASTARTE, a survey has been based on 50 questions and random face-to-face interviews, and administered on the beaches, boats, ports or city centers, with the collection of 1,159 questionnaires, in 6 coastal areas of France, Greece, Norway, Portugal, Spain and Turkey (Goeldner-Gianella et al., 2017; Liotard et al., 2017). Lack of information on tsunamis and non-adapted awareness campaigns are the main explanations. Television seems also to strongly influence the perception, as 90% of the respondents consider that they are the first source of information to understand the tsunami risk. On
the other hand, Cerase et al. (2019) studied differences between citizen's perception and the hazard assessed by the scientific data, and the people's perception on media representations of catastrophic events, with the collection of 1,021 questionnaires including 474 men and 547 women aged between 18 and 95 years, across 138 coastal municipalities of Apulia and Calabria regions (Italy). This survey, based on 27 questions, administered to landline users (833) or mobile phone users (183). In detail, 41 % of the respondents believe that the occurrence of a tsunami appears 'unlikely' in these two regions. 30 % of Apulian
citizens consider their coasts as likely to be hit by a tsunami, whereas the figure for Calabria exceeds 60 %. A possible explanation of this difference is in the frequency and time interval of tsunamis that have occurred in the two regions. These results are consistent with a similar study by Gravina et al. (2019). However, they also observed a low transfer of information and experiences, from generation to generation.





To complete this sample, Cugliari et al. (2022b) recently collected questionnaires in eastern Sicily (614) and in south-northern
Italy (4207), to have 5,842 completed questionnaires that are representative of 8 regions and 37 provinces. 60% of the
respondents declared the Adriatic and nearest seas are 'unlikeness' of a tsunami, whereas 20 % noted the Tyrrhenian Sea, Strait
of Sicily, Sea of Sardinian, or Sardinian Channel. No significant variations are observed in relation to the gender of
respondents, but a higher educational degree is correlated to a higher tsunami risk perception (48.2 % compared to 37.9 % for
those with a lower educational degree). Last but not least, differences are detected between inhabitants of metropolitan cities
and those residing in the municipalities of the respective coasts: 36.3 % of the respondents believing in Rome think a tsunami
may hit their municipality, but 48.2 % for those living on the Tyrrhenian seaside or in Naples. The comparison among cities
shows a higher tsunami risk perception in Reggio Calabria, Catania or Naples, and the authors postulated the history and the
repetitive disruptive natural events, including earthquakes, volcanic eruptions, and tsunamis, may explain such results.

### 2.1.3 But no clear trends on the precise influence of each stimulus worldwide

Stimuli can therefore differently influence the tsunami risk perception. Several studies support strong ground shaking as the
main, or even likely the only sign for deciding to evacuate (Lindell et al., 2015; Buylova et al., 2020). For people located inside
buildings, ground shaking causes them to go outside as they can easily access the additional cues prompting evacuations (Dohi
et al., 2015). However, for far-field events or tsunami-earthquakes, in which the ground shaking is weakly felt, the tsunami
risk cannot be perceived, and ground shaking cannot be an efficient sign (Mori et al., 2007). Although ground shaking is not
the only warning cue that people can notice before the arrival of a tsunami. Significant water withdrawal may also lead people
to evacuate the shore but it may not be sufficient to trigger evacuation by itself (Murakami et al., 2012) and sometimes it does
not happen (Bird and Howes, 2008). Feeling both ground shaking and noticing anormal sea movement can provide additional
cues to the population and start evacuation movement (Mori et al., 2007). Some people therefore declare they seek information,
through neighbors or official websites, before deciding the evacuation stage (Lindell et al., 2015), or want to confirm the
authenticity of safety guidelines (Woody & Ellison 2014). Others want to be sure of the safety of friends and family, especially
during the daily activities, when the family members are separated, whereas the emergency personnel emphasize the
importance of immediate evacuation without helping or looking for other people (Buylova et al., 2020). Others also declare
they collect goods, lock their houses (Goto et al., 2011), or take time to close their activities, so they do not immediately shift
to the evacuation movement stage (Sun & Sun, 2019; Ong et al., 2023). Time is especially crucial: Japan, which has
experienced many large-scale tsunamis, promotes a policy of 'Tsunami Tendenko'. This expression, that can be translated as
'go separately' (Kodama, 2015), instructs the population not to contact family members, or not to wait for tsunami alerts and
confirmation, wherever people find themselves (Goto et al., 2011; Buylova et al., 2020). People can also decide to evacuate
during the late period (Makinoshima et al.,2020), just before the first wave. Additional signs can finally be expected, such as
hearing clashing boats (Gregg et al., 2006), the loud noise similar to a train, a landing plane or an explosion (Gregg et al.,
2006; Bird and Howes, 2008), or even seeing a distant pale water wall (Atwater et al., 1999).





Evaluation of the influence of each stimulus on tsunami perception thus remains a strong glaring gap. People declaring locking houses when they feel ground shaking vary from 14% to 26% (Cerase et al., 2019), but this has reached 52% during the 2014 event (Goto et al., 2020) or 59% during the 2018 event (Harnantyari et al., 2020). During the 2009 Samoa event, 37% declared they were waiting for family news before evacuating, and 24% for neighbors (Lindell, 2015), while 11% evacuated seeing the

neighbors running during the 2011 Japan event (Murakami et al., 2012). In the latter study, they also showed that only 3% evacuated when they see an anormal sea movement, while 21% evacuated feeling the ground shaking. On the other hand, the oldest (up to 65 years old) are more likely to be socially isolated, and can have chronic diseases and limitations in terms of daily activities that hamper their ability to communicate about and respond to a disaster (Malik et al., 2018; Shih et al., 2018). But this is challenged by numerous studies. The surveys by Cox and Kim (2018) or Cong et al. (2021) have for example shown

that the level of preparedness for natural disasters may be similar for the older middle-aged (50-64 years) and even higher than for the younger groups (18-34, 35-49). Compared to high school or below degrees, individuals with a higher educational level seem to be less likely to be prepared than their counterparts, which is explained by the fact they were self-sufficient and more confident about their judgment on perceived probability and consequences of disasters (Hall et al., 2022). Hence, the precise effect of each stimuli remains unclear. And we assume not evaluating many other variables (*e.g.,* past experience, household

emergency preparedness, hazard proximity…), nor cognitive aspects (*e.g.,* self-efficacy, community tenure…).

### 2.2. The alert perception

### 2.2.1 Definitions and main features

In brief words, the alert is a signal sent to the population on an imminent or already ongoing danger or threat, then requiring the application of safety instructions in a very short time duration (Sorensen 2000, Douvinet et al., 2022; EENA, 2023). The

recommended reaction facing a tsunami alert is the rapid evacuation from the hazard zone (Wei & Lindell, 2017). Then, the form and the content of tsunami alerts issued to the general population is important for their correct interpretation. Often confused, the alert differs from a warning, as the latter induces recommendations facing one event that will occur in a few hours. Other potential warnings could include unusual wave formations and currents in both the ocean and estuaries that expose portions of the ocean floor not normally visible, even during low tides (Gregg et al., 2006). The alert perception can thus be

defined as the process of interpreting stimuli (sounds, texts, voices) and understanding the related signification. Alert tools capture the individual's attention with various formats (a light, a sound or a text), to interrupt daily activities (Douvinet et al., 2021). The alert perception also includes the identification of actors who alert them and duration for applying safety measures. One efficient tsunami alert system then should connect: 1) the hazard detection center (implicating meteorologists, geologists and/or industrialists), defined as a 'Tsunami Service Provider' (TSP); 2) the decision-making center that takes the decision to

alert population; 3) the targeted population. The alert perception also includes capability of people to identify any danger, for reducing as possible human and economic losses caused by a disaster (Cain et al., 2021).



### 2.2.2 A tsunami alerting system recently deployed over the French metropolitan coastal areas

To alert of a probable tsunami arriving on coastal areas bordering the French Mediterranean basin, a specific tsunami alerting center was created in 2012: the CENALT (Centre National d'Alerte Tsunami in French). The system consists of a dense monitoring network and six alert centers covering the North East Atlantic and the Mediterranean, forming the NEAMTWS. Since 2016, some framework exercises (involving crisis management actors) have been organized every first Tuesday of each month and municipalities can participate if required. The alerting process is organized in two sequences: a rising alert (from the tsunami detection to information of authorities) and a falling alert (from authorities to the population). Once a tsunamigenic earthquake is detected and confirmed, the CENALT locates the seismic epicenter, evaluates its magnitude and its depth, defines a risk level and transmitted this information to the COGIC (the so-called *Centre Opérationnel de Gestion Interministérielle de Crise* in French) within a legislative duration of 15 minutes. The COGIC, in charge of the falling alert, has a duration of 5 minutes to send alert to the prefectures, and to municipalities which are directly integrated to the COGIC due to high exposure of human issues, or by political choice. These authorities can then alert the population with their available tools.

A national report (Courteaux, 2019) however highlighted several problems in the diffusion of the alerts during trials, which were underlined during recent exercises organized with the COGIC, and surveyed by us. During the World Tsunami Day, on November 5th, 2018, the alerting message arrived in 34 minutes at the Alpes-Maritimes prefecture, and some municipalities playing this exercise even received it after 45 minutes. Since then, the holding of regular exercises has improved the alert transmission times, and avoided multiplying the number of intermediate actors. In November 2021, the alert was transmitted in 15 minutes to the crisis unit and the mayor of the town of Cassis, during a national exercise, and the next exercises followed in 2021 and 2022 confirm this incompressible 15-minute delay.

### 2.2.3 But rare is the precise evaluation of alert perception worldwide

Whereas the trials allow technical tests and the improvement of the information transmission chain, between the different services engaged, the population is rarely involved: falling alerts were never tested including population until now, so nobody knows how people should react in the face of a real tsunami alert in the French Mediterranean basin. Over the last years, limited works of literature have also been carried out on the alert perception. First generalist works have been initiated on the issues for national alerting systems (Sorensen 2000; Rogers & Tsirkunov 2011), but other works are only focused on technical reports and feedback collected after real events (Bopp and Douvinet, 2022). If the organizational aims are identical, the actors involved in issuing alerts neither used the same references nor the same tools, and the population is rapidly out of the scope. Tools in place are influenced by the political context, and crises that occurred in the past contributed either to the transformation or the improvement of the national warning system (Bean et al., 2015). In France, however, if more and more actors become aware of the importance of community involvement and population empowerment, the alerting process remains too vertical to really empower local communities, despite the use of tools that could enable it (Bopp and Douvinet, 2022).



As a consequence, the second glaring gap is the precise evaluation of the alert perception. Different examples can illustrate this: 1) in France, in 2014, only 22% of the population declared identifying the sound related to sirens, but without indication on how they react if sirens are sounding (Douvinet et al., 2021); 2) in The Netherlands, 32% of the population downloaded the national mobile application (EENA, 2023), but without additional information on their usefulness in the situation of dangers or threats. In addition, even if very rare are the studies who analyzed reactions of the population during real alert, data show sometimes significant differences. 39% of the population decided to evacuate hearing the alerting sirens during the 2011 Japan event, but 18% evacuated without waiting alarms (Murakami et al., 2012). Receiving a visual message on media and television might improve the evacuation: 51% of the surveyed Mauritius well received the 2004 tsunami alert on their television for example (Perry, 2007). But electrical or communication networks can be affected by earthquakes (Goto et al., 2011; Murakami et al., 2012). Then, these alerting tools can become insufficient. During the 1960 tsunami event in Hawaii, the siren rang three hours before the tsunami arrived, and the meaning of the signal was not clear to most of the inhabitants (Atwater et al., 1999). If 95% of the population heard sirens, only 32% evacuated (Lachman et al., 1961). During the tsunami on Samoa Islands, in 2009, city bells were also used, so the population were aware that a danger was arriving, but the specific tsunami signal was not used, so the part of the population who knew the signal was waiting to evacuate (Lindell et al., 2015). Even if the signals seem understandable by a great number of people, some of them also do not respect the instructions, due to denial, misunderstanding, other priorities, or due to echo phenomenon, as attested by some residents about the tsunami speakers of Yamada during the 2011 event (Murakami et al., 2012). The alerts can also be disseminated by third parties such as strangers, relatives or neighbors. Yun and Hamada (2015) reported for example that 21% of the responding survivors received verbal tsunami warnings by persons around them.

## 2.3. Research gaps and objectives

Unfortunately, no studies combining tsunami risk and alert perceptions have been previously carried out, and especially in the municipalities along the French Mediterranean basin. Most of the previous studies also returned to the problem of "milling", which is a consistent theme in emergency research (Wood et al. 2015, 2017; Smith et al., 2022): if the recipients are uncertain about the alert they received, or about how they should proceed, they seek additional information, which may delay taking protective actions (Wood et al., 2017; Kim et al., 2022). Even if it is not advisable in an emergency. To bridge this gap, we created and deployed a face-to-face questionnaire, adapted from Liotard et al. (2017), Sun & Sun (2019), Cerase et al. (2019) and the ASTARTE project (2013-2016). The objectives of this survey were fourfold: (a) to estimate the tsunami risk perception in 5 municipalities, characterized by high densities, but in which authorities do not inform on tsunamis using the same strategy; (b) to assess the perception on tools, duration and authorities who deliver alerts; (c) to detect any socio-demographic differences between the surveyed areas; (d) to identify if intentions on perceived risk can be related to the alert perception.

These aims are translated into three research questions:

**RQ1.** Does people's perception in the 5 French coastal cities match with previous studies?



**RQ2.** Can significant differences in the tsunami perception or in the alerting knowledge be explained by demographic or geographic variables? If yes, which one, and can we explain these differences?

**RQ3.** Can we relate the tsunami risk to the alert perception, and if not, why?

Understanding, this tsunami risk perception and alerting knowledge is thus helpful to ensure the confidence of local residents and workers in the authorities who can decide to alert and evacuate. Our results can also be used to drive the communication

strategy the municipalities may develop to increase knowledge on tsunami, and to implement civil protection actions. Outputs can also support the development of the UNESCO Tsunami Ready program, especially in progress in Cannes.

# 3. Methods and data

Data for this study draw from a total series of 750 face-to-face questionnaires, with 150 collected in five French municipalities situated along the Mediterranean coastline. Central for this survey was to validate where we interviewed residents or workers.

For this, we used the tsunami evacuation zones, defined as the flooded areas in which a tsunami may endanger the population. Before discussing the obtained results, the data collection and issues are then explained in this section.

### 3.1. Studied areas

The municipalities in which the surveys have been conducted have been selected for various reasons. First, they are densely-built areas (**Table 1**). With an overall area of around 1,962 ha and a population of 72,435, according to the census population

(INSEE, 2020), Cannes is the most densely populated municipality. Sanary-sur-Mer and Bastia present similar sizes but weaker densities (i.e., 48,296 and 17,173). Six-Fours-Les-Plages has the highest size (2,658 ha) but less density, as Bastia (**Figure 1**). Second, the 5 municipalities present highest densities within the tsunami evacuation zone, even if the number of residents or the size of these zones differ (**Table 1**). The tsunami evacuation zone was validated as *"the zone with an altitude of less than 5 meters and a distance from the sea of less than 200 meters along the river entrances; this distance is extended to 500 meters*

*from the coastal rivers mouth"* (Carles et al., 2023). The deterministic approach was used due to the various simulation models, in terms of resolution, sources used, modeling codes and spatial coverage (BRGM, 2022). The deterministic approach therefore presents several advantages: 1) to have a common regional coverage but with precise resolution; 2) to apply this approach by integrating low-occurrence scenarios (Mw > 8 earthquake in Algeria, for example); 3) to integrate maximum sea level rising, feared with the global warming (Alhamid et al., 2022). The maximum height (5m) remains consistent with the results of the

probabilistic model (PTHA), tested and applied in Italy in the framework of the European project TSUMAPS-NEAM (2018). Third, the choice of these municipalities was motivated by the scientific knowledge accumulated for several years (**Figure 1**). Since 2016, Cannes has organized awareness-raising activities each year, we participated, especially during the International Tsunami Day (the November 5th), and since 2021 Cannes is dealing with UNESCO for the Tsunami Ready label assessment. In Bastia, a communication strategy on the tsunami has been discussed in 2019, and on April 25th 2015, 20 terraces in seaside



restaurants were evacuated by the authorities, who banned traffic on the beachfront for almost an hour, even if it was finally a 'false alert'. In Bandol and Cannes, the evacuation has been planned, while in other surveyed areas, none exist. The objective was then to see if differences could be observed in the tsunami risk perception evaluation, as well as in the alert perception.

**Figure 1: Location of the surveyed areas, with sirens and the evacuation zone mapped in each municipality.** Copyright: the authors



**Table 1: Characteristics of the 5 municipalities located along the French Mediterranean basin.**

| Studied municipalities (abbreviation) | Municipality size (in hectares) | Census Population (habitants) | Tsunami evacuation zone (in hectares) | Residents in the evacuation tsunami zone (habitants) | Density in the tsunami evacuation zone (in hab.ha) | Period of the survey (year, month, day) | Number of collected questionnaires | Final sample |
|---|---|---|---|---|---|---|---|---|
| Bandol (BAN) | 858 | 8,359 | 35 | 939 | 268.2 | 2019/12/09 | 122 | 150 |
| | | | | | | 2020/01/20 | 28 | |
| Bastia (BAS) | 1,938 | 48,296 | 80 | 1,191 | 148.9 | 2019/10/24 | 71 | 150 |
| | | | | | | 2019/10/25 | 79 | |
| Cannes (CAN) | 1,954 | 72,435 | 74 | 4,961 | 181.0 | 2019/11/29 | 124 | 150 |
| | | | | | | 2020/01/23 | 26 | |
| Sanary-sur-Mer (SAN) | 1,924 | 17,173 | 44 | 1,882 | 427.7 | 2020/09/14 | 97 | 150 |
| | | | | | | 2021/06/26 | 53 | |
| Six-Four-Les-Plages (SIX) | 2,658 | 35,339 | 20 | 1,902 | 158.5 | 2020/09/15 | 100 | 150 |
| | | | | | | 2022/09/24 | 50 | |

Four, the studied areas are not covered by the same number and quality of alerting tools: 6 sirens, covering approximately 78% of the residents in an audibility radius of 2km (Douvinet et al., 2021) exist in Cannes in 2019, and 2 sirens are located not sor

far from the highest densely evacuation zone (**Figure 1**). A SMS calling system (*TéléAlert*) also allows potentially informing more than 33,000 people (date obtained in 2018). In the four other municipalities, only 1 siren exists, and they are often located far from the tsunami evacuation zone. In Sanary-sur-Mer and Bandol, a SMS calling system also exists, but we have no idea about the number of registrations. The surveyed areas were finally chosen within the evacuation tsunami zone and in the inner centers, to maximize the number of face-to-face questionnaires.

**3.2. Procedure and survey design**

**3.2.1 Survey procedure**

We invited the residents or workers we met directly in the evacuation tsunami zone to participate in this sample. Once half of the expected questionnaires were collected (75 out of 150), a provisional assessment was made between interviewers, to see if the expected age categories needed to be rebalanced. Due to the containment measures enacted due to the pandemic COVID-

19 in France, and especially the 3 periods (17 March-11 May 2020; 20 October-15 December 2020; 3 April-3 May 2021) when the displacements were strictly forbidden, the survey was carried out during different days (**Table 1**), and takes longer. We therefore decided to maintain the face-to-face process, to avoid disruptions in the procedure, to ensure the presence of the same interviewers to administer questions, and to guarantee we have all answers to the questions at the end of the questionnaire. 86 people started to answer the questionnaire, but they stopped until the end. So, their answers were not retained.

The questionnaire was available in French. On the first page, the participants can see logos of the official partners that support this study, and the logo of the municipality, who gives its agreement in advance. The objectives of the survey were briefly presented to make as better as possible the survey credible. All the respondents were also invited to give their consent to take part in this research, by respecting the GDPR (General Data Protection Regulation) protocol, in place in France since 2015.



The questionnaire was co-designed with the help of psychologist colleagues we invited to participate in March 2019. Five
students were asked to comment on any aspects of the questionnaire in April, 2019. Based on their feedback, a few questions
(Q6, Q9, Q14) and related answers were reworded slightly, to eliminate redundant answers. A list of predefined handsets was
proposed, without using Likert scale or particular response options, that reflect the level of agreement (or disagreement) with
a statement from the respondents, due to the fact that distances between response options cannot be presumed equal based on
the values assigned to the response options. The 5 interviewers who administered the questions were also formed one week
before the first campaign, during one day, to ask questions in the same order and, as best as possible, with the same tonality.

### 3.2.2 Data collection

The face-to-face questionnaire consists of 5 sections and 18 questions, with 7 closed and 11 opened questions (**Appendix 1**).
In Section 1, we invited the respondents to declare (Q1) if they are working and/or living in the area in which we interviewed
them, how long ago (Q2), and where they lived before (Q3). In Section 2, we asked them if natural or industrial risks can occur
in the surveyed area (Q4), without proposing a list of predefined handsets and without mentioning the tsunami risk until then.
If one or several risks were declared, we asked (Q5) how they think they could be alerted for each one, using the classification
proposed by Bopp and Douvinet (2022), by opposing traditional tools (siren, radio, door-to-door, mobile loudspeaker), to other
tools based on numerical (SMS) or modern vectors (social media, internet), or by who (mayors, prefects, etc.), adapted from
Cerase et al. (2019).

In Section 3, we declared to the respondents we are dealing with the tsunami risk, indicating that the Mediterranean coastline
can be confronted with tsunamis caused by earthquakes, landslides (i.e., the event occurred during the Nice airport building in
1979; we do not declared to the respondents at this stage of the questionnaire) or meteorites, and that such events can generate
waves and water levels that may threaten the coasts. No questions were addressed on the term tsunami because no other term
exists in these areas, while the term "maremoto" exists for example in Italy (Cerase et al., 2019). The interviewers question
the respondents with three open questions as follows: Q6. You are on the seafront and suddenly feel the ground shaking, so
what do you do? Q7. You feel the ground shaking on the seafront but you also see an anormal sea movement, so what does
this mean? And what to do (Q8)? These questions were adapted from Dohi et al., (2015) and Lindell et al. (2015). We also
want to check if the ground shaking related to water level movement further triggers evacuation intentions (Murakami et al.,
2012). We then asked the respondents (Q9) if they know the maximum levels that could take inundation during a tsunami,
with 6 closed handsets ('0-1m', '1-2m', '2-5m', '5-10m', 'More than 10m', or 'I don't know'), adapted from Liotard et al.
(2017). We also asked the respondents if (Q10) they can freely describe the consequences that a tsunami could induce over the
municipality in which they were questioned, how they think they might be alerted (Q11) and by who (Q12)? The next question
(Q13) refers to the time duration between a tsunami and its arrival locally. 7 closed handsets were proposed ('Less than 10
minutes', 'between 10 to 30 minutes', 'between 30 to one hour', 'between one to five hours', 'between 5 to 24 hours', 'more
than 24 hours' or 'I don't know'), adapted from Filippini et al. (2020).



In Section 4, we asked the respondents how they should react if they received a SMS (Q14), defined as follows: "Tsunami alert! Move away from the seafront and go high ground" and how long it would take for them to react (Q15). If the respondents declared they may seek information, we asked them how and from who (Q16). Finally, in Section 5, questions concern (Q15) the gender ('male', 'female', 'other' or 'no answer') and (Q16) age group (referring to the grid of the French National Institute

for Statistics and Economic Studies). The 5 interviewers can write a free comment on the final feeling of each respondent (Q17) or their specific reactions (Q18). Once the questionnaire is completed, a discussion is also engaged with each participant, during 15 to 30 minutes, to give them the answers expected in the questionnaire. Such a process was a great and nice occasion to inform them on the tsunami risk along the French Mediterranean basin.

### 3.2.3 Measurements and statistical methods

The 750 face-to-face questionnaires were analyzed using *Sphinx online* and *RStudio*, to allow filtering and application of different, but complementary, statistical analysis. Three approaches were carried out. First, a qualitative analysis was carried out to compare the average values obtained in each site. Figures summarizes results by opposing accepted (with a variation of green color ranges) to negative (with variation in the red color ranges) answers. Second, for each question, the chi-square method $\chi^2$ was used to statistically confirm trends in the collected data we detected. The age, gender and the residency status

(residents or workers) were used as explanatory variables. When the null hypothesis (H0) was rejected (with $p<0.001$), the Cramer index (Ci) was calculated for measuring the strength of the independence relationship. Third, two Multiple Correspondence Analysis (MCA), one referring to the tsunami risk perception (Q4, Q6, Q7, Q8, Q9), another to the alert perception (Q11, Q12, Q13, Q14) were carried out. In addition, two HAC (*i.e.,* Hierarchical Ascendant Classification) were conducted on the coordinates of the two previous MCA, to group the respondents in function on the occurrence of the accepted

(or false) answers. Once classifications were analyzed, we looked at the distribution of the respondents in each of the HAC, and studied if the sites joined, or not, the same "profiles". If yes, a strong relationship between the tsunami risk and the alert perceptions is proven. If not, then, we studied why there is any difference, and followed the discussion in the last section.

### 3.3. Limitations

Methodologically, the face-to-face questionnaires create various data biases. First, this non-probabilistic approach is based on

the availability of participants, their geographical proximity or their willingness to participate in a survey, rather than on well-defined statistical criteria to select and include subjects (Cesare et al., 2019). Unfortunately, this does not guarantee results' significance and generalizability (Etikan et al., 2016). Second, wording or order of questions can play a role in the way the respondent answers (Davis and Venkatesh, 1996). As an example, if we mention the tsunami in Section 3, several people suggest coming back to Section 2, but the 5 interviewers were instructed not to do so. Third, there is commonly a bias between

the intended declarations and the reality. We have already measured this recurring mismatch (Douvinet et al., 2022) between behavioral duties (what individuals declare they know how and what to do) and their behaviors they should have in times of crisis, agreeing with other studies carried out in psychology (Weiss et al., 2011) or in tsunami perception (Cugliairi et al., 2022;



Ong et al., 2023). Four, the period during we collected the 750 answers also introduced biases. In Cannes, a storm induced significant floods on September 2019, 23th (*i.e.,* 6 days before the first survey in this area) and sirens were activated, so this

could affect the results. In addition, the COVID-19 crisis was indicated as a risk for several respondents (Q4). Five, during the survey, it was delicate to ask people to declare their socio-professional situations and incomes, especially as it may decrease the response rate. Several studies showed influence of these factors (Cong et al., 2021; Hall et al., 2022, Cugliari et al., 2022b). Following the previous recommendations (Slovic, 1987; Dominey-Howes and Goff, 2010), the face-to-face questionnaire was however the most suitable method of investigation. This allows to obtain accurate data, as well as to check data quality and

their validity, even if this method takes time and requires human resources in the field experiment to avoid bias during the data collection. As 150 questionnaires were collected in each site, we can also compare the results between the five surveyed areas.

# 4. Main results

After presenting the profile's respondents, the results obtained on the global sample and for each site are presented in the two first subsections: tsunami risk perception, and alert perception. The classifications are compared in the last subsection.

**4.1. Profile's respondents**

Presented in **Table 2** are the descriptive statistics of the 750 respondents. The participants that were interviewed had 53.6% female and 46.4% male, ranging from 60 to 74 years old (29.1%), 45–59 years old (25.7%), 30-34 years old (18.3%), 15–29 years old (16.9%), and the rest (10.0%) were 75 years old and above. 68.1% of the respondents are living in the surveyed areas, while 31.9% are only working in it. As seen in **Table 2**, several differences appear in comparison with the French census

population reported for each municipality. The global sample had an overrepresentation of respondents aging from 15 to 74, whereas the 75 years and above are less represented, whatever the municipality. In detail, the 45-59 are overrepresented in Bandol, Bastia, Cannes, while the 60-74 are overrepresented in Six-Fours-Les-Plages. These differences can be explained by the fact that the census population considers the part of 0-15 years old, while in this survey, no children under the age of 15 years old were interviewed, to respect the French regulation that prohibits interviewing such an age group. Another difficulty

was to interview aged people (Sun and Sun, 2019), to meet them and to convince them to answer the face-to-face questionnaire. 34 of the 86 respondents who stopped the survey were over 75 years old. While these biases are consistent with other questionnaires (Cerase et al., 2019; Ong et al., 2023), their influence is limited as long as it is not so severe that it attenuates the variances of the variables (Lindell & Perry, 2000; Chen et al., 2021).






**Table 2: Samples and census demographics of data collected through the face-to-face questionnaire.**

| | Studied areas (abbreviations) | Bandol (BAN) | Bastia (BAS) | Cannes (CAN) | Sanary-sur-Mer (SAN) | Six-Four-Les-Plages (SIX) | Average in % (* on global sample) |
|---|---|---|---|---|---|---|---|
| Location of respondents | Only for working | 52 | 34 | 99 | 26 | 28 | 31.9 * |
| | Living here | 98 | 116 | 51 | 124 | 122 | 68.1 * |
| Gender | Female (and %) | 68 | 81 | 84 | 85 | 84 | 53.6 * |
| | Men (and %) | 82 | 69 | 66 | 65 | 66 | 46.4 * |
| S - Age groups of respondents in our sample (in %) | 15-29 | 16.7 | 21.3 | 14.0 | 12.0 | 20.7 | 16.9 |
| | 30-44 | 12.0 | 27.3 | 24.0 | 16.7 | 11.3 | 18.3 |
| | 45-59 | 30.7 | 25.3 | 29.3 | 21.3 | 22.0 | 25.7 |
| | 60-74 | 31.3 | 22.7 | 24.0 | 33.3 | 34.0 | 29.1 |
| | > 75 | 9.3 | 3.3 | 8.7 | 16.7 | 12.0 | 10.0 |
| C - Age groups (%) in the census population (INSEE, 2020) | 15-29 | 10.2 | 16.4 | 16.4 | 10.4 | 11.4 | 12.9 |
| | 30-44 | 12.2 | 21.5 | 17.1 | 11.8 | 15.2 | 15.6 |
| | 45-59 | 19.6 | 18.7 | 18.6 | 18.4 | 20.5 | 19.1 |
| | 60-74 | 27.3 | 15.7 | 18.7 | 27.4 | 23.3 | 22.5 |
| | > 75 | 19.9 | 10.0 | 17.1 | 22.1 | 16.7 | 17.1 |
| Differences between sample (S) and census (C) | 15-29 | 6.5 | 4.9 | -2.4 | 1.6 | 9.3 | 3.9 |
| | 30-44 | -0,2 | 5.8 | 6.9 | 4.9 | -3.9 | 2.8 |
| | 45-59 | 11.1 | 6.6 | 10.7 | 2.9 | 1.5 | 6.6 |
| | 60-74 | 4.0 | 7.0 | 5.3 | 5.9 | 10.7 | 6.7 |
| | > 75 | -10.6 | -6.7 | -8.4 | -5.4 | -4.7 | -7.1 |

**4.2. An underestimated tsunami risk perception, but encouraging intentions**

**4.2.1 A common tendency to underestimate the tsunami risk, as other risks**

15.8% of the 750 respondents declared tsunami as a risk (**Figure 2a**), whereas 20.1% declared natural risks (floods, forest fire) and 21.8% technological risks. All these risks are therefore present in each municipality. 20.1% declared that Sanary-sur-Mer is likely to be hit by a tsunami, while 47.3% declared no risks. In Bandol and Cannes, the number of respondents identifying the natural risks are important (51.2% and 50%), but only 13.3% and 16.6% mentioned the tsunami. In Bastia and Six-Fours-Les-Plages, 28.6% and 29.3% identified natural risks but only 13.3% a tsunami. For this question, no statistical relationship exists between the identification of a tsunami or the other risks with the age ($\chi^2 = 2.82$; p>0.57), nor with the gender ($\chi^2 = 0.73$; p>0.39) or the residency status ($\chi^2 = 3.43$; p >0.48).

**4.2.2 The ground shaking as a natural sign for generating protective intentions**

When the 750 respondents imagined they were on the seafront and felt ground shaking (Q6), 13.4% declared going high ground, 11.7% moving away from the seafront and 22.1% leaving. 47.2% of the 750 respondents then declared they might take effective and protective actions. 13.2% declared they shelter themselves in buildings. We can consider this answer as a sufficient protective action, but this is not the official safety instructions in case of a tsunami (**Figure 2b**). In contrast, 18.9% declared do not leaving, 13.8% do not knowing what they do, and the remaining 6.6% join various answers that are as weak (<1.3%) as diverse (*e.g.,* calling for help, calling friends, going to the sea, worrying, getting on a boat, calling schools).



Therefore, if the percentages of respondents declaring they should leave are close between the areas (*e.g,,* ranging from 18.2 to 24.1%), data collected for other intentions are different. 19.3% and 18.7% of the respondents declared going high ground in Bastia and Cannes, but only 5.8% declared this in Six-Fours-Les-Plages. While 19.3% and 16.6%, in Sanary-sur-Mer and Bandol, declared they should move away from the sea, only 7.4% and 4.6% declared it in Cannes and Bastia. On the other hand, 18.1% in Cannes declared they do not know what to do, 17.3% in Bandol, 16.0% in Bastia, but 10.6% in Sanary-sur-Mer, and 8.1% in Six-Fours-Les-Plages. In addition, 24.7% in Bastia declared they do not move, 21.2% in Cannes, 20.5% in Six-Fours-Les-Plages, 19.3% in Bandol, but 8.7% in Sanary-sur-Mer. Finally, the number of respondents who declare protective actions is not correlated with the number of respondents who do not know what to do and who do not leave. As a good example, in Six-Fours-Les-Plages (34.2%), the weakest number of persons who declare leaving is due to the high number of respondents who shelter themselves in a building (25.5%). The chi-square test with the age ($\chi^2 = 53.22$; p <0.001; Ci = 0.13) shows that individuals aged 30-44 are more likely to take shelter, while those aged 45-59 are less likely to do so. Those aged 75 or above have a greater tendency to declare they do not leave, despite also indicating they should move away from the sea. Therefore, the analysis of residuals indicates that the frequencies of the responses 'I go to higher ground' and 'I leave' are relatively infrequent compared to the other responses.

### 4.2.3 Greater protective intentions when ground shaking is associated to an anormal sea movement

If the respondents imagined they are on the seafront, but see also an abnormal movement of the sea level (Q7), 46.5% of the global sample declare identifying a tsunami, 14.5% refer to a meteorological event, an earthquake or a danger, whereas 38.9% do not know what is happening. In detail, strong differences are detected: 68% of the respondents identified a tsunami in Six-Fours-Les-Plages, 62% in Sanary-sur-Mer, 52.6% in Cannes (**Figure 2c**), 40% in Bastia, but only 22% in Bandol. The chi-square test with the location ($\chi^2 = 123.46$; p <0.001; Ci = 0.28) confirms the high differences of no answers in Bandol, and the high differences of the number of respondents in Six-Fours-Les-Plages who identified the tsunami. However, no other relationship with age, gender or residency status is detected. We also asked the respondents how they should react (Q8). 29.1% of the global sample declare going high ground and 36.2% leaving. So, this question cumulates (65.2%) more positive answers than Q6. Higher number of respondents declare moving away from the seafront in Sanary-sur-Mer (20.7%) or Six-Fours-Les-Plages (16,7%), rather than for the 3 other sites (**Figure 2c**). In contrast, the number of respondents who declare sheltering strongly decreases (ranging from 0 to 2%). 17.1% declared they do not know how they react, 8.3% do not leave and the other answers joined 8.3%. The chi-square test with the location ($\chi^2 = 79.91$; p <0.001; Ci = 0.16) confirms weak differences of respondents who move away from the seafront or do not leave in Bastia, and the higher values for the respondents who move away from the seafront in Sanary-sur-Mer. The chi-square test with the age ($\chi^2 = 49.5$; p <0.001; Ci = 0.12) confirms that the 60-74 often declared moving away from the sea, whereas the 75 and over are those who do not leave, but also who report the most sheltering. Nevertheless, we do not observe other statistical relationships regarding gender or the residency status.





**Figure 2: Data collected for addressing the tsunami risk perception (Part 1).** Copyright: the authors



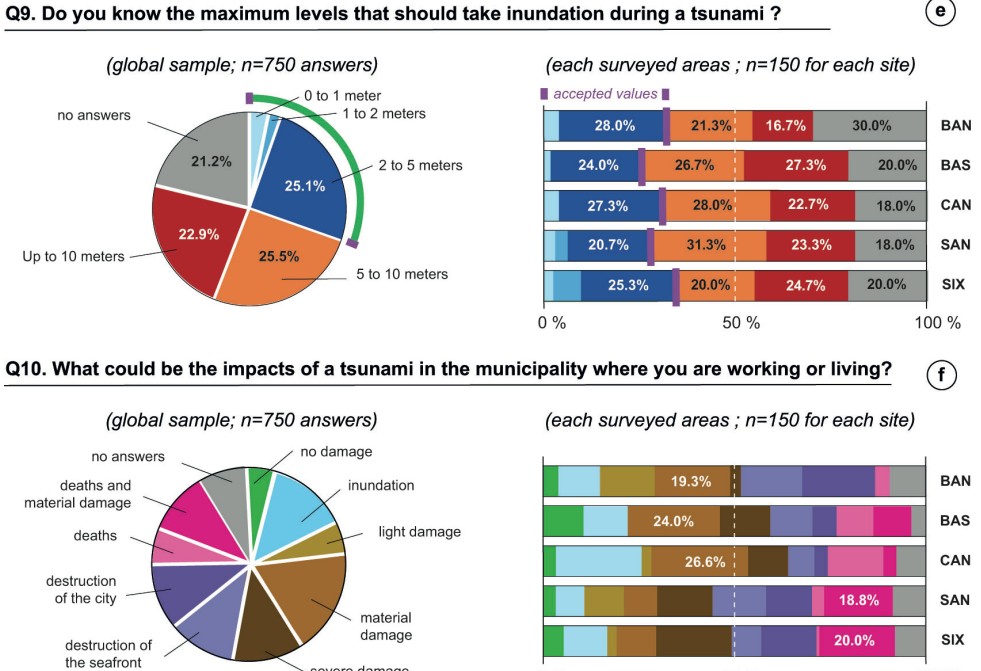

**Figure 2: Data collected for addressing the tsunami risk perception (Part 2).** Copyright: the authors

Comparing Q8 and Q6 shows strong variations. The association between the anormal sea movement and the ground shaking strongly increases the number of respondents going high ground (+24.2% in Six-Fours-Les-Plages, +20.6% in Bandol, +15.4% in Sanary-sur-Mer, +12% in Bastia), except in Cannes (+5.3%). The number of respondents who declare leaving is slightly higher in response to the two signals (*e.g.,* +6.6% in Cannes or +4% in Bastia for example). The number of respondents who declare leaving from the seafront extremely varies, in negative (-9.3% in Bandol) but also in positive (+6.5% in Six-Fours-Les-Plages) proportions. On the other hand, the number of respondents who do not leave extremely decreases (-21.4% in Bastia, -12.6% in Bandol), while the number of respondents who do not know how they react is nearest. Finally, the cumulative percentage of respondents who take protective actions increase in various proportions (ranging from +34.5% in Six-Fours-Les-Plages to +9.8% in Cannes), but exceed more than 60% in the 5 surveyed sites (**Figure 2d**).

### 4.2.4 Misconceptions on the inundation water heights and probable damage due to a tsunami

Presented in **Figure 2e** are the results we analyzed for the height of waters the respondents declared (Q9). 25.1% of the global sample declared that a tsunami could generate values ranging between 2 to 5 meters. Higher values were obtained in Bandol (28%) and Cannes (27.3%), while the weakest value was collected in Sanary-sur-Mer (20.7%). If we accept that answers can include heights ranging from 0 to 5 meters, the accumulated rate equals to 30.4% (e.g., 35.3% obtained in Six-Fours-Les-Plages, but only 26% in Bastia). The number of respondents that declared probable heights ranging from 5 to 10 meters (25.5%)



or exceeding 10 meters (22.9%) is important as these two modalities cumulate a total percentage of 48.4%. During the survey, many respondents declared to the interviewers that a major tsunami may occur, evocating the 2004 event, media and the photos they saw over the events observed overwide. The number of respondents who do not answer is highest (30.2%) in Bandol. The

chi-square test with the location ($\chi^2$ = 49.5; p <0.001; Ci = 0.13) confirms the high differences in the number of respondents who declared a maximum height of 1-2m in Six-Fours-Les-Plages, and the differences in the number of 'no answers' in Bandol. However, we do not find any other significant statistical associations between gender, age group or the residency status.

Related to the previous question, the respondents were also invited to imagine the consequences of a tsunami in the area where they are working or living (Q10). 4 groups emerge: on one hand, some respondents indicated that tsunamis do not induce

damage, or just small inundation or light damage. These three responses cumulate 24.1%, but varies from 23.3.% (in Bastia) to 30.6% in Bandol, and 28.6% in Cannes (**Figure 2f**). Material and severe damage constitutes the second category, with accumulated percentages ranging between 22% (in Bandol) to 38% in Bastia. Destruction (i.e., of the seafront or of the city) is the third item. The accumulated percentage is strongly diverse, ranging from 10% (in Cannes) to 34% (in Bandol). On the final hand, tsunamis should unfortunately provoke deaths, related or not to material damage. The number of respondents who

declared this is higher in Sanary (18.8%) and Six-Four-Les-Plages (20.0%) rather than in the other surveyed areas. In Cannes, 14% declared tsunamis can induce deaths, without severe material damage. The chi-square test with the age ($\chi^2$ = 56.58; p <0.001; Ci = 0.17) confirms that the youngest (15-44 years) anticipated at least some material damages, compared to the oldest, who expect small or none effect on the coastline. In addition, post-hoc analysis of the standardized results indicates that the observed frequencies for the 75 years or more who anticipated no effects were significantly lower than expected.

**4.3. An overestimated alert perception, mixing positive intentions and erroneous beliefs**

**4.3.1 A common confusion between decision-makers in the alert and warning services**

When we asked who are the actors to alert of a tsunami (Q12), 25.1% declared mayors and 4.3% prefectures, so these two accepted answers only equal 29.4% (**Figure 3a**). In contrast, 9.9% falsely think they should be alerted by safety services, 7.2% by televisions, and 1.6% directly by the Tsunami Service Provider (TSP). 45.5% do not know who should alert them, and 5.9%

think an alert should be never sent. Differences exist at local scales, as statistically attested by the chi-square test with the location of respondents ($\chi^2$ = 98.1; p <0.001; Ci = 0.25). While 40% of the respondents identified mayors in Sanary-sur-Mer, only 15.3% in Bastia. 63% of the 150 respondents in Bandol did not know who should alert them, but only 37.3% in Sanary-sur-Mer. In this municipality, the mayors and safety services are two answers more currently quoted. No statistical relations exist however between the identification of actors with the age ($\chi^2$ = 4.54; p>0.27), nor with the gender ($\chi^2$ = 1.45; p>0.36).

**4.3.2 Satisfying perceived time duration between tsunami arrivals and alerts**

55.7% of the respondents estimated the time of a tsunami arrival (Q13) to be less than 1 hour after receiving an official alert, with one third (21.6%) estimating it to be less than 10 minutes, and 18.1% from 10 to 30 minutes. 2.8% precise they cannot



give a precise duration by stating that it depends on the location of earthquake or of the event generating the tsunami. Then we can consider this result satisfying regarding the current state-of-the-art. In Sanary-sur-Mer, the final rate even equals 63.3%.

In contrast, 26% of the respondents were unable to answer this question, 7.8% gave time exceeding 5h, and in Bastia, 9.8% of the respondents declared a time ranging from 5 to 24 hours (**Figure 3b**). The chi-square test with the location ($\chi^2 = 70.9$; p <0.001; Ci = 0.15) confirms the high differences we describe before. The chi-square test with the age ($\chi^2 = 74.81$; p <0.001; Ci = 0.16) shows that the 15-29 are more numerous to declare 1 to 5h, 45-59 a duration of less than 10 minutes. The 15-44 are also those who least report not knowing, unlike the 75 years and above.


**Q12. Who may alert you in the situation of a tsunami?**                                                                    (a)

*(global sample; n=750 answers)*                    *(each surveyed areas ; n=150 for each site)*

**Q13. How long between a tsunami and its arrival locally?**                                                               (b)

*(global sample; n=750 answers)*                    *(each surveyed areas ; n=150 for each site)*

**Q14. How should you react receiving one Alert SMS?**                                                                      (c)

*(global sample; n=750 answers)*                    *(each surveyed areas ; n=150 for each site)*

**Figure 3: Data collected for addressing the alert perception (Part 1).** Copyright: the authors



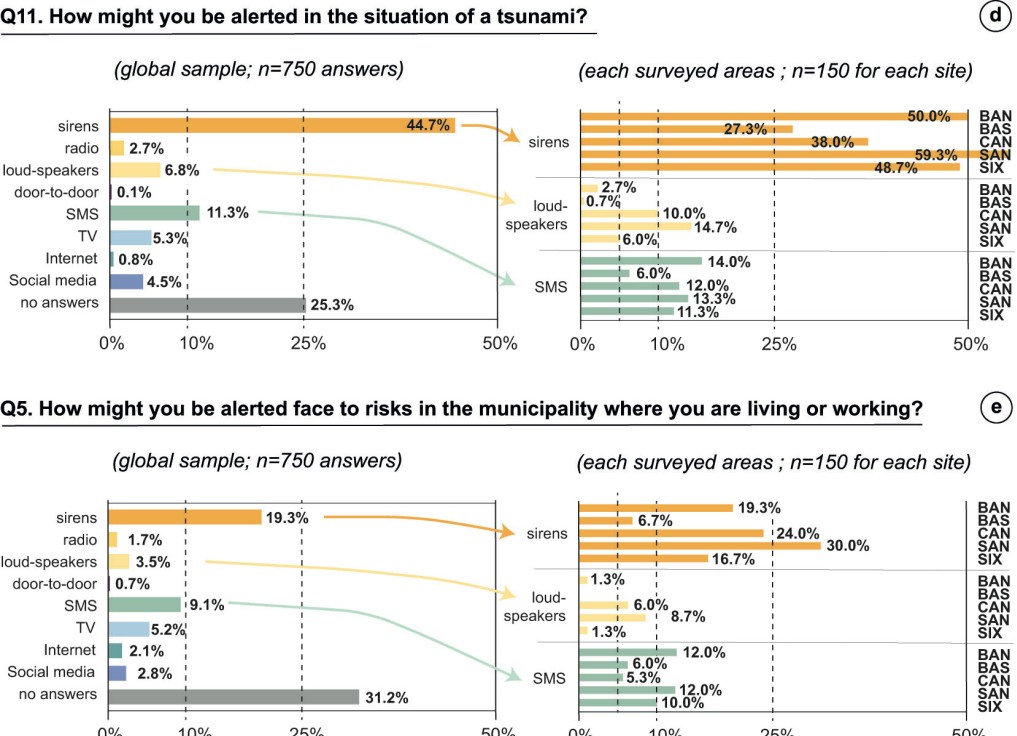

**Figure 3: Data collected for addressing the alert perception (Part 2).** Copyright: the authors


### 4.3.3 Equally satisfying intentions if people imagined receiving a Tsunami Alert SMS

In addition, we asked the respondents (Q14) how they think they would react upon receiving a SMS mentioning *"Tsunami alert! Move away from the seafront and go high ground"*. 55.7 % declared they immediately walked high ground with values ranging from 49.3% in Bastia to 68.7% in Bandol (Figure 3c). However, before leaving, 10.1% declared they take time to
inform the family members, 7.5% to inform people located around, 7.5% to seek information to check the credibility to SMS, 4.7% to gather their belongings, 4.7% to look for their car, 3.1% to pick up their children's. All these answers represent 37.5%. In addition, 4% of the 750 respondents declared they remain not in danger (8.7% in Bastia), 1.7% do not know what they should do (4.7% in Six-Fours-Les-Plages) and 1.7% shelter into a building (3.3% in Cannes). This short SMS is thus not sufficient to generate protective intentions in the situation of a tsunami and is coherent with recent studies (Cain et al., 2021;
Smith et al., 2022). The term 'going high ground' was not totally understood by the overall respondents, some of them reported going upstairs (4.7%) or using their car (1.3%). The chi-square test with the residency status ($\chi^2 = 13.89$; $p < 0.001$; $Ci = 0.17$) reveals that respondents who work in the surveyed areas are less likely to evacuate immediately, compared to those who live in. However, the chi-square test cannot be used for variables such as gender, age group and residency status.





### 4.3.4 But a recurrent myth of sirens, inducing erroneous beliefs

Other results confirm that sirens remain falsely perceived as vectors of alert in the situation of a tsunami. First, 93.2% of the global sample declared (Q11) they might be alerted of a tsunami and surprisingly, whereas they could indicate several tools, 91.6% mentioned only one. 44.7% declared they might be alerted by sirens, but without any relation with the number of sirens really existing in each municipality. The higher rate in Sanary-sur-Mer (59.3%) differ from those analyzed in Cannes (38%) or Bastia (27.3%). 11.3% of the respondents secondly declared they surely might be alerted by SMS, with the higher values in

Bandol (14%), Cannes (12%), Sanary-sur-Mer (13.3%), where a SMS call system exists. In contrast, values in Six-Fours-Les-Plages (10.1%) and Bastia (5.9%) do not refer to any existing tool. Thirdly, loud-speakers obtain the higher number of answers in Sanary-sur-Mer (14.7%) and Cannes (9.1%), where such tools are deployed. Values in the three cities are weaker, probably due to the absence of such tools (**Figure 3d**). Television (5.3%), social media (4.5%) and radio (2.7%) cumulated a very low response rate. On the other hand, 25.3% declared they do not know how they might be alerted, even as high as 42.7% in Bastia.

Interestingly, the chi-square test with the location ($\chi^2$ = 130.5; p <0.001; Ci = 0.28) confirms the high differences in Bastia (in the overrepresentation of 'no answers' and 'social media', but also in the underrepresentation of 'sirens' or 'loud-speakers'), in Sanary (high number of 'loud-speakers') and in Six-Fours-Les-Plages (overrepresentation of 'no alert').

Results obtained when we previously asked the respondents to think about the alert in the situation of risks (Q5) show other results (**Figure 3e**). Only 19.3% of the 750 respondents identified sirens, with higher variations between the five sites (30.0% in Sanary-sur-Mer, whereas only 6.7% in Bastia). 9.1% mentioned SMS and except in Cannes, the values are quite similar

than for Q11. Only 3.5% identified loud-speakers, with values slightly higher for Cannes and Sanary. On the other hand, 40.7% were not able to give a precise answer to this question, while this was lower in Q5. Such differences can be explained by the fact that we have given more details about the tsunami risk when we asked the question Q11, allowing the respondents to be probably more sensitive to alerting tools. However, the proportion of respondents who were aware of the existence of sirens

for alerts (Q5) is very close to the figures (22%) given at national level (Douvinet et al., 2021). The chi-square test with the location ($\chi^2$ = 154.2; p <0.001; Ci = 0.31) confirms the highest number of 'social media' and 'meteorological services' in Bastia, but no statistical relations are proven for the other variables.

### 4.4. Lack of reciprocal relationships between tsunami risk and alert perceptions

In the first MCA, of the total inertia of 1, 29.3% is accounted for by F1 (e.g., the first axis), 20.4% by F2, 19.1% by F3, 16.9%

by F4 and 14.3% by F5. Cumulatively, 49.7% of the overall inertia is accounted for by the first 2 axes. The negative answers of the respondents (Q7) when they felt ground-shaking and saw the anormal sea movement (Contr. = 0.19), and when (Q9) they estimated the water heights of a possible tsunami (Contr.= 0.11) contributes the most to F1. The negative answers (Q9) are also underlined in F2, with a clear association to the lack of identification of the tsunami (Contr.= 0.08) in Q4. Respondents who positively answer to Q8 (Contr.= 0.18) but who rejected answers to Q9 (Contr.= 0.17) are grouped in F3. In F4,

respondents who mentioned tsunamis as a risk in Q4 (Contr.= 0.25) and who gave positive answers to Q6, Q7 and Q8 are included. The HAC (i.e., Hierarchical Ascendant Classification) realized using the MCA coordinates, and the analysis of the



dendrogram, enables us to identify 4 classes (**Figure 4a**). Group 1 joins most of the respondents who mostly provided incorrect responses to Q8 (140 persons out of a global sample of 255) and to Q9 (141 out of 517), and who did not mention the term "tsunami" (143 out of the 630). Group 2 consists of respondents who gave a good response to Q9 (with 86% of the overall number of respondents who correctly answered to this question), but also who did not mention the tsunami as a risk (196 people out of 630). 44% of this class did not also provide a correct answer to Q8. Group 3 joins a large part of the respondents who proffered an accurate response to Q8 (291 out of 489), and abstained from mentioning the term "tsunami" (291 out of 631) and yielded an incorrect response to Q9 (291 out of 522). Finally, Group 4 solely consists of respondents who well identified the tsunami at the beginning of the questionnaire (119 out of 119). In this Group 4, 75% of the respondents also demonstrated proficiency in answering Q8, while 56% displayed proficiency in answering Q7. As a result, among Group 4, the respondents who included the term tsunami in their response displayed a greater tendency to identify the arrival of a tsunami and to apply the appropriate safety measures.

The total inertia of the second MCA (**Figure 4b**) gives following results: F1 accounts for 30,9%, F2 for 25,5%, F3 for 22,5% and F4 for 21%. The cumulative contribution of the first 2 axes amounts to 56,4% of the total inertia. The incorrect responses of respondents when we asked to cite the alert systems dedicated in case of a tsunami (Q11) as well as the duration before tsunami arrival (Q13) contributes the most to F1 (with respectively Contr.= 0.2 and 0.17). The respondents who failed to implement instructions receiving the SMS (Q14, Contr.= 0.36) and who did not identify the actors involved in the tsunami alert process (Q11, Contr.= 0.29) explained the second axis. F3 derived from the contribution to Q13 (positive answers: Contr.= 0.23; negative answers: Contr. = 0.27), whereas Q14 contributes mostly to F4 (for positive answers: Contr.= 0.29, and negative answers: Contr. = 0.31). The classification performed with this second HAC also allows the identification of 4 classes. Group 1 consists of respondents who answered incorrectly to Q14 (198 persons out of the global sample size of 287) and Q12 (198 out of 526). Additionally, a significant part of 59% who failed to provide correct answers to Q11 (116 out of 366), while 56% were found to answer Q13 incorrectly (109 out of 353). This group joins the respondents who have a gentle understanding of the alerts. Group 2 only involves respondents who gave inaccurate replies to Q13 (167 out of 397) and Q12 (149 out of 526), while they should apply instructions receiving the SMS (149 out of 463). Group 3 includes most of the respondents who successfully responded to Q13 (179 out of 397) and Q14 (179 out of 463) but also those who did not answer Q12 accurately (179 out of 526). Ultimately, Group 4 is composed of 100% of individuals who demonstrated precise knowledge in answering Q12 (224 out of 224), as well as 59% of those who accurately responded to Q11 (133 out of 383). However, whilst the most of respondents in Group 4 have precise knowledge by answering both Q11 and 12 correctly, it is worth noting that a substantial proportion of 40,6% also provided an erroneous response to Q11 (91 out of 367). While the first classification allows the identification of groups in which respondents have positive intentions (Class 4), in contrast, this second highlights a group (Class 1) that has a very little understanding of alert, both on actors, procedures, timeframes and protective actions.

The distribution of the 150 respondents in each group, from the two classifications, finally gives a synthetic overview for each surveyed areas, we can then compare, and early findings clearly appear (**Figure 4c**). The co-occurrence of low perception for each respondent seems infrequent, even if the distribution in Six-Fours-Les-Plages stands out from the rest. Indeed, the number



of respondents who exhibit high perceived levels (i.e., grouped in Class 4 in the two classifications) is 24% higher than the low perception match-up. Nevertheless, these observations need to be discussed, by considering that only 6.8% of the surveyed individuals (51 out of 750) possess the highest level of perception for our two variables.

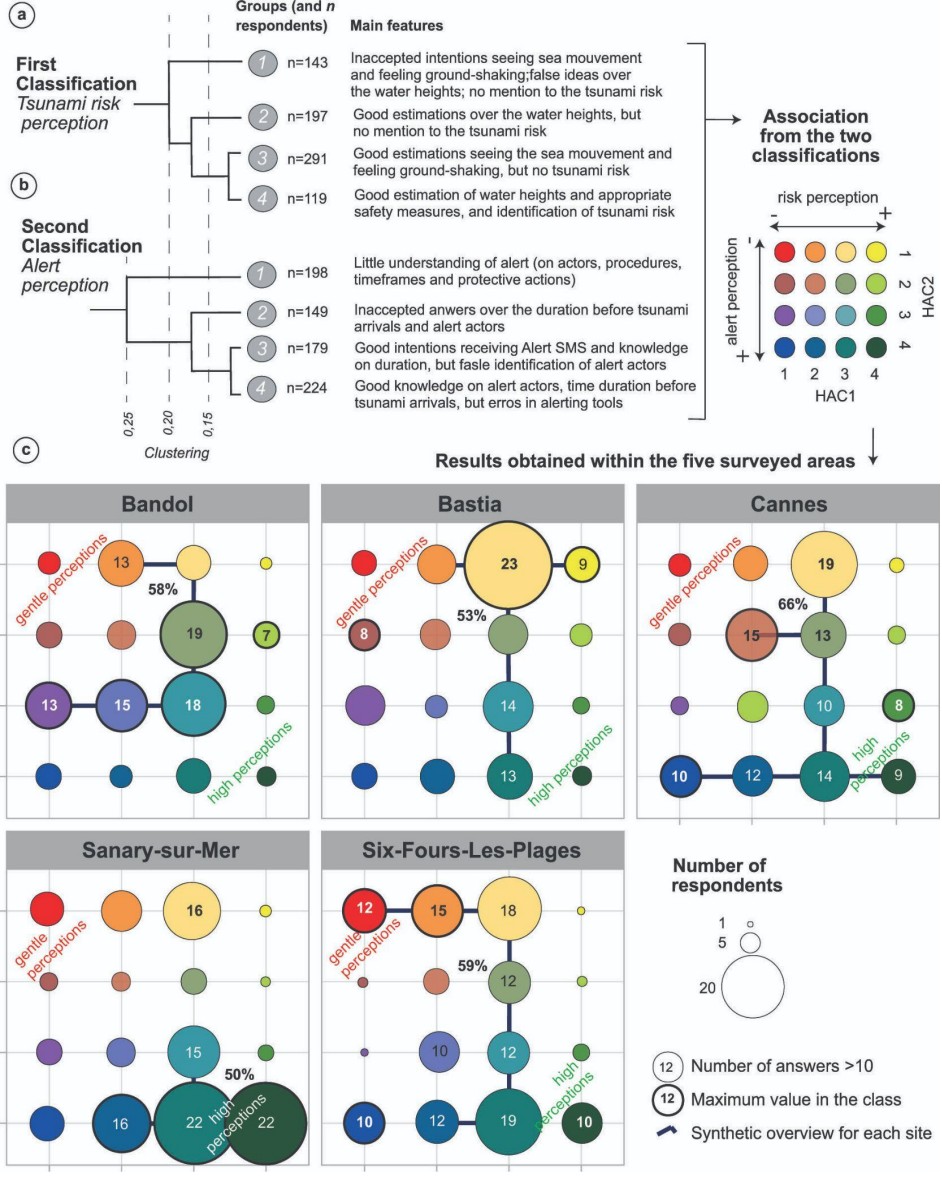

600

**Figure 4: Differences between the surveyed areas combining the MCA and HAC classifications** Copyright: the authors

In Bandol, the overview reveals an overrepresentation of profiles with an average level of knowledge toward both variables, as 60% of respondents overlap in Groups 2 and 3. Cannes bears the strongest resemblance to Bandol, even though there is a



difference in the number of respondents possessing a more accurate sense of the alertness, which is approximately twice as much in Cannes (**Figure 4c**). In Bastia, the number of highly perceptions for the respondents is the lowest among all the surveyed cities. Considering the global sample, the deviations from the mean across the profile combinations are consistently negative for those with the sharpness perception. In Six-Fours-Les-Plages and Sanary-sur-Mer, a scarce presence of moderately perspectives profiles, while notable differences set them apart from each other. Six-Fours-Les-Plages counts an equal number with low and high perception profile, whereas in Sanary-sur-Mer, the balance shifts markedly in favor of the respondents endowed with a heightened perception. With 63 (out of 150), Sanary-sur-Mer emerges as a model with a significant proportion of respondents exhibiting a high level of perception regarding tsunami risk and alert. Last but not least, the 75 years and above are overrepresented in Groups 1 and 2, characterized by weak perceptions. It could be a problem, due to the exposure to the disaster and low capacity of aging people (Arimura et al., 2020; Hall et al., 2022), but also because most of them lived alone, took no action to respond to a tsunami, but are also less likely to be associated with disaster preparedness (Sun & Sun, 2019).

## 5. Discussion and recommendations

Our results confirm that the public's understanding of a tsunami appears to be biased by images of catastrophic stand-alone events, rather than a more realistic idea of events that have various ranges of intensity (Rafiana et al., 2022). However, the public's perception in the five surveyed areas seems to indicate several variations in comparison with the current state-of-the-art (**RQ1**). First, if most of the respondents present the common tendency to ignore that tsunamis could affect them (Ong et al., 2023), the tsunami likeness is weaker (15.8%) than in the previous study (24.5%; Liotard et al., 2017), or in nearest areas like Italy (ranging from 30 to 41% for Cerase et al., 2019). This might be explained by the lack of major historical events that occurred during the last decades, and the awareness-actions and communication actions do not induce major differences in our results. Second, it is interesting to note that a large part of respondents answered positively regarding the duration before a tsunami arrives (55.7% from 0 to 1h), but incorrectly answered on the alert senders (29.4%) and on the water heights (30.4% from 0 to 5 meters). Although some respondents understand and seek information to know what a tsunami is (Lindell et al., 2015), and know what they should do and preventive actions if a tsunami happens (Ong et al., 2021), other information seems thus to be not useful for them. Third, the content of the SMS alert message does not guarantee the appropriate reaction and can push people into a more dangerous situation.: only 55.4% of the respondents declare going high ground, while this is mentioned in the textual message. In contrast, and hopefully, a large part of the respondents indicated that they would be willing to have positive intentions feeling ground-shaking on the seafront (46.5%), and even more if they also saw an anormal sea movement (64%). This confirmed a relatively high perception when natural signals are perceived (Arce et al., 2017). Our estimations are clearly higher than in other risk perception studies (for example, between 14 to 26% in Cerase et al., 2019), and interestingly, values are nearest from those observed after real events (Goto et al., 2020; Harnantyari et al., 2020).



Surprisingly, the level of the alert perception is seen to be statistically not related to the perceived tsunami risk (**RQ3**). Whereas official authorities are identified by 29.2% of the respondents, the potential of sirens in the face of a tsunami is overestimated, as 44.7% declared they could be alerted by such tools. 93.2% of the respondents also think they could be alerted, while no official alert may arrive in the situation of a tsunami generated from the Ligurian sea, which could arrive in less than 7 minutes according to numerical simulations (Filippini et al., 2020). These results illustrate the myth of sirens, existing for many years

in France (Douvinet et al., 2021). In brief words, this refers to Aesop's fable (Sawada et al., 2022). Due to the trials (e.g., the sirens are tested each Wednesday per month for more than 60 years) and the current communication (e.g. the authorities currently inform only on these tools), the population think they could be alerted by these tools wherever it may be. Therefore, only 2,248 municipalities (out of a number of 33,545) are equipped with sirens in France and they cover less than 28% of the residents with an audibility radius of 1km (Douvinet et al., 2021).

Furthermore, a relevant finding is that differences exist due to the age of the respondents, more than the residency status and gender (**RQ2**). As people get older, they seem more likely to react passively to the risk of a tsunami (Arimura et al, 2020). In addition, higher aging groups often tend to recognize subjective norms more than younger (Terumoto et al., 2022). In this study, we confirmed that the 75 years and above underestimate risk and alerts, confirming their vulnerability. The youngest have higher perceptions, and it is encouraging as they represent the future generations (Hall et al., 2022). However, this is in

contrast with a recent study in Cascadia that found that age was not significantly associated with earthquake and tsunami risk perception (Chen et al., 2021). On the other hand, gender or residency status do not play a key role, while females seem to be more likely to believe that they are susceptible to a tsunami in other studies (Hall et al., 2022).

Evidence from this survey is consistent with what happened on 21 July 2017 in Greece: a small tsunami was generated by a low magnitude (6.6) earthquake from the island of Kos and the nearby coastal city of Bodrum (Turkey), with run-up elevations

as high as 2 m (Yalçiner et al., 2017). On that occasion, surveillance cameras on the Kos waterfront captured the way people were reacting to sea level anomalies: they were seemingly calm and curious to see the water inundating quaysides. They were shooting pictures and videos with their smartphones instead of fleeing, thus emphasizing that the risk posed by small tsunamis was almost completely ignored. This behavior has also been noticed in Sydney during the 2010 tsunami where hundreds of people waited on beaches (Dominey-Howes and Goff, 2010), or during the 2016 Japanese tsunami where people filmed the

tsunami entering into rivers (Suppasri et al., 2017).

This sample finally questions the tsunami evacuation from the hazard zone (Wei & Lindell, 2017; Hall et al., 2022), defined as the single effective response that can lead to safety (Goto et al., 2012). In this survey, 7.8% and 18.9% of the respondents respectively declare they shelter or they do not leave if they feel ground shaking on the seafront, and 9.8% do not leave even if they see an anormal sea movement. Respecting the current legislation in France, we have not considered these answers as

positive. However, if people go upstairs and follow signposts into the buildings, these actions sound good, and the vertical evacuation could then be included in plans (Rafiana et al., 2022). By this way, as planning vertical evacuation is essential for low-lying areas subject to the risk of local tsunamis, we need to involve designating buildings and to clearly communicate on



which buildings can be used for this (Hall et al., 2022). And a systematic evacuation could be not needed in these areas, as water heights are evaluated from 2 to 3 meters, and with a maximum of 5 meters.

## 6. Conclusion

Slightly fewer than half of our study respondents were susceptible to having protective actions when they feel ground shaking, but more than two-thirds declared moving or going high ground if they also see an anormal sea movement. Thus, these two natural signs generate more reactions but it suggests a need for educational interventions and communication campaigns to increase tsunami perception. On the other hand, the potential of sirens or SMS is overestimated. This confirms the current myth of sirens in France, but it also suggests the need for training and communication actions to explain the limits of these historical tools. At the end of this study, even if alert and tsunami risk are connected, no reciprocal relationships are statistically detected in the surveyed areas.

By combining the tsunami risk and alert perceptions, we can also suggest ways the municipalities can further communicate and inform on tsunamis. This study is struggling to be sustained (Rafliana et al., 2022), but efforts are in progress. Evacuation areas and recommendations, location of memorials throughout the city, photographs of past events, and a summary of the types of evacuation signs are possible to locate in the city. In Cannes, the evacuation itineraries are physically marked along several roads and equipped by many loud-speakers. Trials or current exercises may improve the perception, but actions should be combined and repeated to increase both the tsunami and alert perceptions. In Cannes, the tsunami risk perception is slightly higher than in other areas, but reciprocity with alerts is missing. Garnier and Lahournat (2022) have well showed how Japanese stone monuments, representing elements of both tangible and intangible culture for the population, reflect a desire to commemorate and transmit past events' memory to future generations. These findings should be adapted in France, to highlight the importance of transmission between generations regarding tsunami, and could be useful for designing effective information and communications activities about tsunami risk reduction (Sutton et al., 2020). Memory and commemorations of past events are relevant for the development of risk strategies and to increase population resilience.

**Data availability.** The dataset generated for the present study is not publicly available because the questionnaire has not yet been administered in all French coastal areas, and the data are being further analyzed by the research team. However, data are available from the corresponding author on reasonable request.

**Author contributions.** All Authors contributed to the early stages of the manuscript by individual contributions from their respective research fields. Major contributions are listed as follows: Early compilation of text: JD; NC, PF, MP, Introduction: JD, PF, NC, MP; Literature review: JD, PF, NC, MP; Methods and data: JD, NC, PF, MP; Results: JD, PF, NC, MP; Discussions: JD, MP, PF, NC; internal review: all authors.



**Competing interests.** The author(s) declared they do not have potential conflicts of interests with respect to the research, authorship, and/or publication of this article.

**Disclaimer.** This paper does not necessarily represent DPC's official opinion and policies.

**Publisher's note.** Copernicus Publications remains neutral with regard to jurisdictional claims in published maps and institutional affiliations.

**Acknowledgements.** We sincerely address our acknowledgements to the respondents, the 10 students who helped us to build the face-to-face questionnaire, and the 2 English native speakers who re-read the text. We also thank the anonymous reviewers

for helpful suggestions that improved the manuscript.

**Financial support.** This article was based upon works carried out within the TASOMA project, supported by the INSHS (i.e., Institut National des Sciences Humaines et Sociales) and the IRD (i.e., Institut de Recherche et de Développement) during two years (2019-2021). Results are available here: https://storymaps.arcgis.com/stories/caabb106dd684f67ab758e3ec14f2ad5

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
