# Peer review of "Tsunami hazard perception and knowledge of alert: early findings in Five Municipalities along the French Mediterranean coastlines"

_EGUsphere, 2023_

## Referee Comment (RC2)

[referee-annotated manuscript omitted]

---

## Author Response (AR1)

**Tsunami hazard perception and knowledge of alert: early findings in Five Municipalities along the French Mediterranean coastlines**

5     Author responses

**Referre #1**

The paper is well written, clear in its objectives and effective in its presentation of results. As acknowledged by the authors themselves, the sample used has some significant limitations, but these have been sufficiently discussed and adressed. The

10     main value of the paper is that it provides important data on a relatively little-studied area such as the Mediterranean coasts of France and sheds light on the limitations of the warning channels that can be used for overcoming the so-called last mile issues. The results of the MCA could perhaps have been more clearly explained, but the reviewer acknowledges that this is a subjective assessment and that the paper can therefore be published without further changes. OK, thanks

15     **Reviewer 2**

Most of the remarks indicated in the pdf. file have been included in the new version of the paper (V2), in red color

The manuscript surveys the tsunami risk and tsunami alert perception of the population living or working in five areas on the French coast. The surveyed area is indeed defined within the perimeter of the tsunami evacuation areas. Four of the locations

20     studied are in the south of France and one location is in Corsica. The five areas face the Mediterranean Sea on different sides. The issues addressed in the manuscript are essential to develop virtuous interventions and tsunami risk mitigation policies aimed at the local population and tourists as well as to implement more effective hazard and uncertainty communication tools. The study takes an international perspective and increases the data availability on tsunami risk perception and tsunami alert perception in the NEAMTWS area. The addressed issues are also of multilevel interest: local, national and European.

25

However, the manuscript has many shortcomings that affect various aspects.

A) conceptual gaps, Tsunami hazard perception and knowledge of alert have been defined, as the state-of-the-art has been rewritten, with more explanations and references, as suggested in the pdf. file

B) methodological shortcomings, Sure, several points have been clarified

30     C) shortcomings in data analysis and output presentation Improved

D) inaccuracies and errors of form. The paper has been corrected and proofread by a English native speaker

Specifically, some references cited in the text are not included in the bibliography. Further bibliography on risk perception and tsunami risk perception is recommended, paying particular attention to the complexity of the constructs and how they are addressed. All the references have been checked with Zotero, and many references have been also added.

35

L35 - there are references to specific data without cited bibliography

Yes, you're right

L64 and others - there are references to the manuscript by Carles et al., 2023 which is published in French. It is recommended to include other references rather than referring to a single native-language text, especially when dealing with important

40 concepts or validation of flood zones with respective parameters. Only five cited references are written in French

L68 - There is intense debate concerning the definition of "natural risks". It is suggested to inform about it. However, it is preferable to use 'natural hazards'. Yes, you're right

L71-94-101 etc. there are important generalised shortcomings on risk perception concept. This also emerges as a contradiction in terms in e.g. L94 and 95. The paper deals specifically with tsunami risk and alert perception, therefore, risk perception must

45 be well structured. The state-of-the art has been completely reformulated.

The authors also survey the perception (a sociologically complex and irreducible concept; Slovic, 1980) by considering only stimuli perceived through the senses. This is a theoretical/conceptual error. Therefore, the theoretical focus is shifted from perception to sensation, a psychological concept, see e.g. Graham and Ratoosh, 1962. Yes, you're right

Major methodological and data analysis discrepancies emerge. Unfortunately, the appendix is not available to provide a direct

50 comparison with the questionnaire administered to the respondents. However, the questions described in section 3.2.2 do not match the data presented in the graphs and commented in section 4.2.3. Some of the questions have been reformulated and do not accurately reflect the original text. Confusion between questions Q7 and Q8 where part of the text has been removed, losing its meaning. Appendix 1 (questionnaire) and the new figure (Figure 2) aim to further explain data collection.

55 Furthermore, the authors extracted categories from open questions (e.g. Q8 and Q10) without explaining the methodological procedure. The categories are not mutually exclusive and the data presented in this way are not statistically robust. To carry out such an operation, it is essential to describe the procedure, the instrument used and to present the categories.

Sure, this is not easy to read, so we better describe the procedure and instrument used, and the profile's respondents in the method and data section

60 Since the authors have very interesting data and in-depth interviews, I suggest that the authors review the entire manuscript because, as mentioned above, it is of relevant importance and submit it once the shortcomings have been fixed.

Yes, you're right. We have done it, and we hope this version will suit to all your relevant remarks.